# Identifying Early Risk Factors for Postoperative Pulmonary Complications in Cardiac Surgery Patients

**DOI:** 10.3390/medicina60091398

**Published:** 2024-08-26

**Authors:** Kaspars Setlers, Anastasija Jurcenko, Baiba Arklina, Ligita Zvaigzne, Olegs Sabelnikovs, Peteris Stradins, Eva Strike

**Affiliations:** 1Department of Cardiovascular Anesthesia and Intensive Care, Pauls Stradins Clinical University Hospital, LV-1002 Riga, Latvia; 2Department of Anesthesiology, Riga Stradins University, LV-1007 Riga, Latvia; 3Faculty of Medicine, Riga Stradins University, LV-1007 Riga, Latvia; 4Institute of Radiology, Pauls Stradins Clinical University Hospital, LV-1002 Riga, Latvia; 5Department of Intensive Care, Pauls Stradins Clinical University Hospital, LV-1002 Riga, Latvia; 6Department of Cardiac Surgery, Pauls Stradins Clinical University Hospital, LV-1002 Riga, Latvia; 7Department of Surgery, Riga Stradins University, LV-1007 Riga, Latvia

**Keywords:** postoperative pulmonary complications (PPCs), cardiac surgery, pleural effusion, risk factors, hypoalbuminemia

## Abstract

*Background and Objectives*: Postoperative pulmonary complications (PPCs) are common in patients who undergo cardiac surgery and are widely acknowledged as significant contributors to increased morbidity, mortality rates, prolonged hospital stays, and healthcare costs. Clinical manifestations of PPCs can vary from mild to severe symptoms, with different radiological findings and varying incidence. Detecting early signs and identifying influencing factors of PPCs is essential to prevent patients from further complications. Our study aimed to determine the frequency, types, and risk factors significant for each PPC on the first postoperative day. The main goal of this study was to identify the incidence of pleural effusion (right-sided, left-sided, or bilateral), atelectasis, pulmonary edema, and pneumothorax as well as detect specific factors related to its development. *Materials and Methods*: This study was a retrospective single-center trial. It involved 314 adult patients scheduled for elective open-heart surgery under CPB. *Results*: Of the 314 patients reviewed, 42% developed PPCs within 12 h post-surgery. Up to 60.6% experienced one PPC, while 35.6% developed two PPCs. Pleural effusion was the most frequently observed complication in 89 patients. Left-sided effusion was the most common, presenting in 45 cases. Regression analysis showed a significant association between left-sided pleural effusion development and moderate hypoalbuminemia. Valve surgery was associated with reduced risk for left-sided effusion. Independent parameters for bilateral effusion include increased urine output and longer ICU stays. Higher BMI was inversely related to the risk of pulmonary edema. *Conclusions*: At least one PPC developed in almost half of the patients. Left-sided pleural effusion was the most common PPC, with hypoalbuminemia as a risk factor for effusion development. Atelectasis was the second most common. Bilateral effusion was the third most common PPC, significantly related to increased urine output. BMI was an independent risk factor for pulmonary edema development.

## 1. Introduction

Pulmonary complications are common in patients who undergo cardiac surgery and are widely acknowledged as significant contributors to increased morbidity, mortality rates, prolonged hospital stays, and healthcare costs [1,2]. The most common PPCs after cardiopulmonary bypass (CPB) are atelectasis and pleural effusion, followed by infection, pneumothorax, and lung edema [3,4].

During the CPB, lung metabolic demand depends on blood flow from the bronchial arteries. In addition, bronchial arterial flow on bypass paradoxically decreases, contributing to worsening low-flow ischemia, which normalizes after pulmonary arterial clamping ends [5,6]. A period of ischemia followed by a reperfusion reaction initiated by CPB elicits a systemic inflammatory cascade, contributing to increased arteriolar resistance, pulmonary hypertension, and fluid shift to the interstitial space, resulting in pulmonary edema and pleural effusion [5,7,8].

In addition to general anesthesia, CPB, and the loss of spontaneous breathing, cardiac surgery induces significant changes in lung function. These changes include chest compression during surgery, thoracic deformation, internal thoracic artery dissection, potential diaphragm nerve paralysis, and poor postoperative pain management, all of which can contribute to the development of atelectasis [4,9,10,11]. Patients prone to PPC often have a limited homeostatic reserve due to patient-related factors such as chronic heart and pulmonary diseases, multiple comorbidities, and older age [12,13].

Clinical manifestations of PPCs can vary from mild to severe symptoms with different radiological findings and varying incidence. To better understand the pathogenesis and promptly prevent further PPCs, detecting early signs and identifying influencing factors of PPCs is essential to prevent patients from further complications. Our study aimed to determine the frequency, types, and risk factors significant for each PPC on the first postoperative day.

## 2. Materials and Methods

### 2.1. Design

This study was constituted a retrospective single-center trial conducted at the Center of Cardiac Surgery within the Pauls Stradins Clinical University Hospital, Riga, Latvia, from January 2023 to July 2023. The study received approval from the Riga Stradins University medical ethics committee under reference number 2-PEK-4/66/2023, dated 15 December 2022, and individual consent for this retrospective analysis was waived.

### 2.2. Participants

Participants were adult patients scheduled for elective open-heart surgery under CPB. The exclusion criteria were patients younger than 18, cardiac surgery without CPB, or emergency surgery.

Demographic data and clinical characteristics of patients were obtained from the patient medical records. Demographic information included age, gender, body mass index, left ventricular ejection fraction, Cardiac Operative Risk Evaluation (EuroScore2), and smoking history. Comorbidities included coronary heart disease, arterial hypertension, atrial fibrillation, diabetes, bronchial asthma, chronic obstructive pulmonary disease, and congestive heart failure class by NYHA.

Laboratory tests and blood gas analysis were performed before and after surgery. The results included pO_2_, difference in Ht level, total plasma protein, and serum albumin level. Normal albumin was defined as ≥35 g/L, mild as 30–35 g/L, moderate as 25–30 g/L, and severe hypoalbuminemia as <25 g/L. Data collected during surgery included the type of surgery, cross-clamp and CPB time, CPB priming volume, cardioplegia volume, and blood transfusions. Data collected after surgery included the time to extubation as well as the length of stay in the intensive care unit (ICU).

A chest radiograph was performed 12 h after surgery, and the findings were analyzed.

Patients were mechanically ventilated according to the local protocol using pressure-regulated volume-controlled mode, FiO_2_—0.6, 6 mL/kg tidal volume, and PEEP set at 5 mmHg. Before extubation, patients were ventilated in pressure support mode. After extubation, oxygen was administered via a non-rebreathing face mask at a flow rate of 10 L/min.

The main goal of this study was to identify the incidence of pleural effusion (right-sided, left-sided, or bilateral), atelectasis, lung edema, and pneumothorax and detect the specific factors related to every pulmonary complication.

### 2.3. PPC Definitions

Postoperative pulmonary complications were defined according to published guidelines by the European Joint Taskforce for perioperative clinical outcome definitions. Pleural effusion was described on a chest X-ray with blunting of costophrenic angle, loss of sharp silhouette of the ipsilateral hemidiaphragm in an upright position, displacement of adjacent anatomical structures, or (in supine position) hazy opacity in one hemithorax with preserved vascular shadows. Atelectasis was defined as lung opacification with mediastinal shift, hilum, or hemidiaphragm shift towards the affected area, with compensatory hyperinflation in adjacent non-atelectatic lungs. Pneumothorax was defined as air in the pleural space with no vascular bed rounding the visceral pleura. Lung edema was defined as vascular redistribution, septal lines, interlobular septal thickening, peribronchial cuffing, bilateral opacities, and air bronchogram [14].

### 2.4. Statistical Data Analysis

Data are expressed as mean ± standard deviation or median (interquartile range) as appropriate. The Shapiro–Wilk test was used for the evaluation of normality of the distribution. In order to evaluate the association of risk factors with PPC appearance after cardiac surgery with CPB, we analyzed differences between PPC type groups and non-certain PPC type with univariate analyses (for comparisons between groups, the Mann–Whitney U test was used or, when appropriate, the two-sample *t*-test; Chi-squared test was used to evaluate the categorical prognostic factors). We employed multivariate analysis through binary logistic regression to pinpoint independent risk factors for each specific subtype of PPCs. All statistical analyses were performed by using IBM SPSS Statistics 28 program. A *p*-value < 0.05 was considered to be statistically significant.

## 3. Results

Analysis included a cohort of 314 patients who underwent cardiac surgery with CPB. Of all the patients, 195 (62.1%) were men and 119 (37.9%) women. The mean age of the study group was 66.17 ± 11.01 years, indicating a predominantly elderly population. Out of the 314 patients reviewed, 42% (n = 132) developed PPCs within 12 h post-surgery. Up to 60.6% (n = 80) experienced one PPC, while 35.6% (n = 42) developed two PPCs, and 3.7% (n = 4) encountered more than two PPCs (Figure 1).

### 3.1. Pleural Effusion

Pleural effusion was the most frequently observed complication following cardiac surgery. Pleural effusion was observed in 89 patients (89/314; 28.3%) (with 62 exhibiting it as an isolated PPC and 27 in combination with other PPCs). Left-sided effusion was the most common, presenting in 45 cases (in 31 patients as isolated PPC and in 14 patients as combination with other PPCs). Bilateral was the second most prevalent form of pleural effusion, occurring in 34 patients (22 patients as an isolated PPC and 7 patients within the context of multiple PPC). Right-sided effusion was the least common, occurring in 10 patients (9 patients with isolated pleural effusion and 1 patient with a combination of other PPCs) (Figure 2).

### 3.2. Left-Side Pleural Effusion

Patients presenting with a left-side effusion had a significantly lower ejection fraction (median = 55, [IQR] 48–60) vs. no left-side pleural effusion (median = 59, IQR 50–61), U = 3267, z = −2.108, *p* = 0.035, r = 0.138, longer CPB time (median = 108, [IQR] 78–121.5) vs. (median = 89, IQR 72–107), U = 7339.5, z = 2.576, *p* = 0.010, r = 0.146) and higher bypass priming volume (median = 1250.0, [IQR] 1200.0–1550.0) vs. (median = 1250.0, IQR 1050.0–1450.0), U = 7008.5, z = 2.006, *p* = 0.045, r = 0.114). Decreased serum albumin level 6 h after surgery significantly impacted left-side pleural effusion development (median = 35, [IQR] 33–37) vs. (median = 36, IQR 34–38), U = 4264.5, z = −2.243, *p* = 0.025, V = 0.145). Moderate hypoalbuminemia increased the risk of left-sided pleural effusions after surgery by threefold (OR 2.84; 95% CI, 1.37–5.87; *p* = 0.005). The observed proportions of patients with valve surgery who developed left-sided pleural effusion significantly differ from those without left-sided effusion vs. patients who did not undergo valve surgery, χ^2^(1) = 13.93, *p* = 0.001, Cramer’s V = 0.224 (Appendix A).

### 3.3. Bilateral Pleural Effusion

There was a statistically significant difference in age distribution between the bilateral effusion group (median = 64, [IQR] 55.75–70) vs. the non-pleural effusion group (median = 68, IQR 62–74), U = 3456.5, z = −2.5, *p* = 0.012, r = 0.146. The surgery day urine output in the ICU was significantly higher in the bilateral effusion group (median = 3750.0, [IQR] 2625.0–4625.0) vs. the non-pleural effusion group (median = 2850.0, IQR 2300.0–3700.0), *p* = 0.010, r = 0.146 (Appendix A). 

### 3.4. Atelectasis

Atelectasis was identified as the second most prevalent PPC. It was observed in 44 patients (44/314; 14.0%), occurring as an isolated PPC in 25 and concomitantly with other PPCs in 19. The incidence of atelectasis was significantly higher among women χ^2^(1) = 5.934, *p* = 0.015, and Cramer’s V = 0.138 (low effect size).

### 3.5. Pulmonary Edema

Pulmonary edema was identified as the third most frequent PPC, occurring in 25 (25/314; 7.9%) patients (11 patients as an isolated PPC and 14 patients within the context of a combination of PPCs). There was a statistically significant connection between a lower BMI and a higher incidence of lung edema (median = 25.3, [IQR] 23.42–28.5) vs. (median = 29.4, IQR 26.3–32.45), U = 1897.0, z = −3.844, *p* = 0.001. Those who received blood transfusion had a higher incidence of pulmonary edema vs. the group without transfusion χ^2^(1) = 8.9, *p* = 0.029, Cramer’s V = 0.119 (low effect size) (Appendix A).

### 3.6. Pneumothorax

Pneumothorax was the least frequent PPC, occurring in nine patients—four as isolated PPC and five combined with other PPCs.

To further explore the risk factors that could influence the development of PPC types, we advanced from univariate to multivariable regression analysis. This transition was essential to determining each factor’s independent predictive value.

Multivariate regression analysis showed a significant association between left-sided pleural effusion development and moderate hypoalbuminemia after surgery (OR 2.84; 95% CI, 1.37–5.87; *p* = 0.005). Meanwhile, valve surgery was associated with reduced risk for left-sided PE (OR 0.205; 95% CI, 0.086–0.486; *p* = 0.001) (Table 1).

Independent parameters for bilateral pleural effusion include increased urine output (OR 1.00; 95% CI, 1.00–1.00; *p* = 0.002) and longer ICU stays (OR 1.41; 95% CI, 1.02–1.90; *p* = 0.032) (Table 2).

Higher BMI is inversely related to the risk of pulmonary edema (OR 0.87; 95% CI, 0.769–0.972; *p* = 0.015) (Table 3).

After conducting a multivariate regression analysis on right-sided pleural effusion, atelectasis, and pneumothorax, we identified no independent risk factors.

## 4. Discussion

The aim of our study was to determine the incidence and frequency of various types of PPC after heart surgery and to identify possible specific risk factors for each PPC. We did a retrospective analysis of patients with various types of open-heart surgeries. Our findings show that PPCs remain a frequent problem after heart surgery, as 42% of the patients developed at least one PPC post-surgery, and that left-sided pleural effusion followed by atelectasis were shown to be the most frequent PPCs. The multivariate regression identified factors that match those of earlier studies, such as urine output, days in ICU, BMI, and type of surgery. We also identified rarely expressed modifiable risk factors, such as serum albumin level, that play a role in left-sided pleural effusion development.

The incidence of PPCs after cardiac surgery was close to that presented by other studies, at around 50%, compared to 42% within 12 h post-surgery presented in our study. In Refs. [4,15], 19% exhibited pleural effusion as an isolated PPC and 8% showed it in combination with other PPCs. The total incidence of any pleural effusion in our study was 28%, compared to other studies, where it varies from 10% to 40% [16]. However, considering the various possible mechanisms for pleural effusion development, we also categorized effusion by type (left-sided, right-sided, and bilateral), in contrast to most other studies where isolated left-sided effusion was shown to be the most frequent (51%) among pleural effusion, followed by bilateral with 32%.

Left-sided pleural effusion is usually caused by operative trauma and is typically hemorrhagic and neutrophil predominant [3,17]. Most studies favor the theory that internal mammary artery harvesting is associated with a higher prevalence of pleural effusion [18,19]. Our study showed that valve surgery was associated with reduced risk for left-sided pleural effusion compared to other surgery types. Still, the role of surgery type in effusion development has been controversial. Labidi et al. compared 2892 CS patients and showed that valve replacement was more strongly associated with postoperative pleural effusions than CABG [20], leaving us to think about pleural effusion multifactorial and overlapping etiology. Pleural effusion can also occur secondarily to surgery due to fluid overload and postoperative fluid management, depending on the patients’ inflammatory state in early postoperative period [4].

Our data and the patterns found therein corroborate those from earlier studies, showing a connection between a more prolonged duration of CPB and PPC [21,22]. Univariate data analyses showed that longer CPB time was associated with more frequent left-sided pleural effusion development (median 108 vs. 89 min). CPB induces complement activation, leukocyte activation, and the release of many inflammatory mediators, including oxygen-free radicals, arachidonic acid metabolites, cytokines, platelet-activating factor, nitric oxide, and endothelin [23]. This condition may lead to increased fluid extravasation and pleural effusion development. In addition, a longer CPB can lead to increased hemodilution. Several factors, including dilution secondary to fluid resuscitation, increased catabolism, reduction in synthesis, blood loss, CPB, and redistribution secondary to altered vascular permeability by inflammation, may cause a decrease in serum albumin concentration [5,8,24]. Variations in albumin levels can affect colloid oncotic pressure (COP), disrupting the balance between the hydrostatic and oncotic pressures in pulmonary capillaries. This imbalance in oncotic pressure promotes fluid transfer from the blood vessels into the interstitial and pleural spaces, possibly causing pleural effusion [25,26]. Albumin is a major plasma protein contributing significantly to maintaining plasma oncotic pressure. In total, 21/45 (46.6%) patients with an isolated left-sided effusion had mild to moderate hypoalbuminemia 6 h after surgery. Decreased serum albumin levels 6 h after surgery significantly impacted the left-side pleural effusion development, and moderate hypoalbuminemia raised the risk of left-sided pleural effusions by three times after surgery. Our multivariate regression analysis showed a significant association between left-sided pleural effusion development and postoperative hypoalbuminemia (≤35 g/L). Our study’s results align with a recent study by Liu et al., where serum albumin levels <40 g/L proved to be independent risk factors for PPCs [27]. Priming CPB circuits with human albumin has shown its benefit on COP and extravascular lung water, suggesting a possible hypoalbuminemia effect on pleural effusion development [8]. We believe the lack of statistically significant impact of hypoalbuminemia on right-sided and bilateral effusion development could be attributed to the relatively small sample size in those groups.

Bilateral pleural effusion was the second most prevalent form of PE and occurred in 34/314 (10.8%) of all cases. Our regression analysis showed a significant association between bilateral effusion and increased urine output. Heart failure resulting in left ventricular dysfunction and increased hydrostatic pressure in pulmonary circulation can lead to the leakage of fluid into the interstitium, causing lung edema and pleural effusion [28]. Although our study did not measure diuretic usage, we hypothesize that patients with postoperative heart failure may receive additional diuretics to increase urine output. In our research, the ejection fraction (EF) before surgery did not significantly affect pleural effusion development, but data about the EF after surgery were not analyzed. Our conclusions are restricted to the EF before surgery, although other studies show that heart failure is a risk factor for pleural effusion development [3,17].

In our study, bilateral pleural effusion was associated with increased time in the ICU. Although our effusion stratification into types (left-sided, right-sided, and bilateral) differs from most other study types, our results are consistent with prior research, which found that postoperative pleural effusion increases patients’ time in the ICU due to acute respiratory failure [3,27]. Increased time on the ventilator and the subsequent need for thoracocentesis are poor prognostic signs [29,30].

Patients with low BMI undergoing cardiac surgery are at a higher risk for fluid accumulation [31]. In contrast to most earlier studies, we found a relationship between BMI and lung edema development. This can be explained by possible fluid overload, where patients with a lower BMI might be at a higher risk of lung edema and ARDS development. The Berlin Definition of ARDS was first published in 2012, where ARDS was defined as bilateral opacities on chest imaging that were not fully explained by effusions and lobar/lung collapse or nodules, and not fully explained by cardiac failure or fluid overload [32], although ARDS identification and interpretation for patients after CPB is still debated despite the update in the ARDS definition [33].

Atelectasis is a common cause of hypoxemia and impaired gas exchange after cardiac surgery and was identified in 44/314 (14.3%) of all the analyzed patients. Other authors suggest an incidence of 16–72% [4,34,35]. Our regression analysis did not identify any independent risk factor for atelectasis development. However, various factors have been related to atelectasis, such as the patient’s respiratory function before surgery, the type of surgery, pain management, anesthesia protocol, the use of blood products, diaphragmatic dysfunction, and mechanical lung ventilation [3,4,36,37]. Recent studies have shown that an open lung strategy, together with recruitment maneuvers, low tidal volumes, and a high PEEP, should be considered to reduce atelectasis [4,9]. Appropriate pain management by reducing opioid usage and increasing the use of nerve blocks could be beneficial in reducing postoperative atelectasis development [10,11].

### Limitations

This study has several limitations. First, as a single-center trial with a relatively small sample size, the generalizability of our results may be restricted. The findings might not apply to other patient populations or healthcare settings. Second, although most patients did not have postoperative drains placed in the pleural cavity, our study does not definitively exclude the possibility of pleural content evacuation through drains in those who did. This underscores the need to consider the potential impact of drainage procedures on pleural effusion dynamics and possibly even greater pleural effusion incidence.

## 5. Conclusions

At least one PPC developed in almost half of the patients 12 h after surgery. Left-sided PE was the most common PPC, with hypoalbuminemia as an independent risk factor for pleural effusion development. Atelectasis was the second most common complication, with no independent risk factors identified. Bilateral effusion was the third most common PPC, significantly related to increased urine output. BMI was an independent risk factor for pulmonary edema development.

Many risk factors contribute to developing PPCs, and clinicians must be aware of both the non-modifiable and modifiable factors to identify at-risk individuals and deliver optimal care.

## Figures and Tables

**Figure 1 medicina-60-01398-f001:**
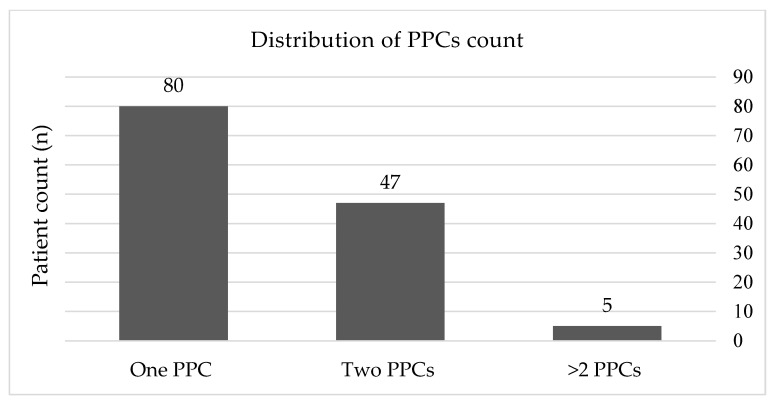
Distribution of PPC count (total number and percentage of count of pulmonary complications). PPCs, postoperative pulmonary complications.

**Figure 2 medicina-60-01398-f002:**
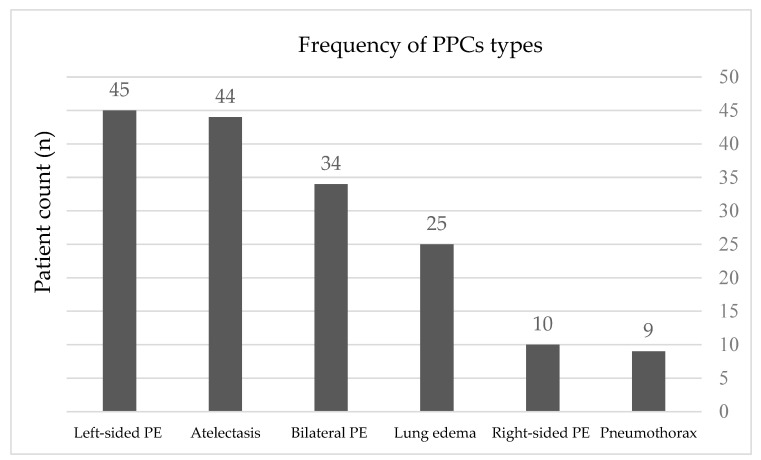
Frequency of PPCs (total number of types of pulmonary complications). Some patients could have several PPCs. PPCs, postoperative pulmonary complications; PE, pleural effusion.

**Table 1 medicina-60-01398-t001:** Multivariate Analysis of Risk Factors for Left-sided Pleural Effusion.

Outcomes	OR	95% CI	*p* Value
EF (%)	0.98	0.98–1.02	0.552
CPB time, min	1.01	0.99–1.016	0.250
CPB priming, mL	1.00	1.00–1.01	0.337
Hypoalbuminemia			
mild (30–35 g/L)	2.84	1.37–5.868	0.005
moderate (25–30 g/L)	4.28	0.397–46.20	0.230
Valve surgery	0.205	0.086–0.486	0.001

Abbreviations: EF, ejection fraction; CPB, cardiopulmonary bypass; SA, serum albumin.

**Table 2 medicina-60-01398-t002:** Multivariate Analysis of Risk Factors for Two-sided Pleural Effusion.

Outcomes	OR	95% CI	*p* Value
Age > 65	0.531	0.27–1.29	0.164
CHD	0.544	0.22–1.22	0.182
Urine output (first 24 h)	1.00	1.00–1.001	0.002
Days at ICU	1.41	1.02–1.90	0.032

Abbreviations: CHD, chronic heart disease; ICU, intensive care unit.

**Table 3 medicina-60-01398-t003:** Multivariate Analysis of Risk Factors for Pulmonary Edema.

Outcomes	OR	95% CI	*p* Value
BMI	0.87	0.769–0.972	0.015
Hypertension	0.42	0.153–1.137	0.098
Intraoperative blood product transfusion	1.63	0.42–5.38	0.420
Postoperative blood product transfusion Hypoalbuminemia	1.184	0.410–3.419	0.755
mild (30–35 g/L)	2.81	0.94–8.40	0.064
moderate (25–30 g/L)	2.96	0.23–37.27	0.401
Days in ICU	1.12	0.925–1.338	0.258

Abbreviations: BMI, body mass index; SA, serum albumin; ICU, intensive care unit.

## Data Availability

Data are unavailable due to privacy or ethical restrictions.

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
