# Peer review of "Identifying Early Risk Factors for Postoperative Pulmonary Complications in Cardiac Surgery Patients"

_medicina, 2024, doi:10.3390/medicina60091398_

Round 1
Reviewer 1 Report
Comments and Suggestions for Authors
Dear Authors,
thank you for the opportunity to review your work. You studied an interesting and significant topic for physicians dealing with post operative cardiosurgery patients.
I would, however, recommend to revise the text and more clearly emphasize the timepoint of the complications you are focusing on: complications occuring on the first postOP day/first 24 hours after the procedure or immediately after the procedure (especially in the ˝Conclusion˝ chapter).
I would recommend to revise the ˝Limitations˝ chapter and clarify the tex in lines: 294-298.
I would also recommend to avoid the usage of abbreviation PE for pleural effusion, as it is widely used in the context of pulmonary embolism.
With kind regards.
Author Response
Thank you very much for taking the time to review our manuscript. Your comments were very useful in improving our research paper. Please find the detailed responses below and corrections in the re-submitted files.
Comments 1: I would, however, recommend to revise the text and more clearly emphasize the timepoint of the complications you are focusing on: complications occuring on the first postOP day/first 24 hours after the procedure or immediately after the procedure (especially in the ˝Conclusion˝ chapter).
Response 2: Agree. I have clarified the timepoint of PPC in the conclusion section[line 309] and also modified (first postOP day/first 24 hours) to “surgery day” [line 167 and Table A4].
Comments 2: I would recommend to revise the ˝Limitations˝ chapter and clarify the text in lines: 294-298.
Response 2: Agree. I have modified the “Limitations” paragraph by explaining the possible impact of pleural drainage on pleural effusion identification and development. Change can be found – in the limitations paragraph, lines 300 - 305].
Comments 3: I would also recommend to avoid the usage of abbreviation PE for pleural effusion, as it is widely used in the context of pulmonary embolism.
Response 3: Agree. I have changed all “PE” abbreviations in the text.
Reviewer 2 Report
Comments and Suggestions for Authors
Line 24. I think that 312 should be replaced by 314.
Line 25. CPB is not explained, you are probably referring to cardiopulmonary bypass.
Line 122. You mention the alpha value as a parameter of statistical significance, but you report the p value to the results (Tables 5-7). You should clarify this aspect.
Please standardize the reporting of the values in Figure 1 and Figure 2, use either numerical values or numbers and percentages.
Lines 231 – 232. You mention the use of the internal mammary for CABG. Do you not use this technique and if so, how many patients have had this procedure?
The cause of hypoalbuminemia is not discussed... it is possible to appear in those with a more severe inflammatory reaction after CPB or due to a large volume of administered fluids.
You don't mention what were the average CPB time and also the relationship between CPB duration and complications, being known that the longer the CPB duration is, the greater the inflammatory response will be. Please enter the CPB duration values, assess the association between CPB time and PPC and after that redo the statistics. How did you monitor the volume status in the postoperative period? (Vexus score?)
Line 254. lack instead of Lack
Lines 263 – 264. Are you sure that EF was not also measured postoperatively? It is hard for me to believe that in a reference university center the cardiac function of patients undergoing cardiac interventions is not monitored in the postoperative period. If this is really true, you better not mention it in the final text.
In general, this study, although it has a large number of included patients and has a good statistical data processing, does not bring new information regarding PPC compared to other studies present in the literature for many years.

Comments on the Quality of English LanguageNo comment
Author Response
Thank you very much for taking the time to review our manuscript. Your comments were very useful in improving our research paper. Please find the detailed responses below and corrections in the re-submitted files.
Comments 1: Line 24. I think that 312 should be replaced by 314.
Response 1: Agree. I have revised this point.
Comments 2: Line 25. CPB is not explained, you are probably referring to cardiopulmonary bypass.
Response 2: Agree. I have revised this point.
Comments 3: Line 122. You mention the alpha value as a parameter of statistical significance, but you report the p value to the results (Tables 5-7). You should clarify this aspect.
Response 3: Thank you for pointing this out. P-values were used as a parameter of statistical significance. I have modified this point. (line 123)
Comments 4: Please standardize the reporting of the values in Figure 1 and Figure 2, use either numerical values or numbers and percentages.
Response 4: Thank you. I have modified Figure 1 and Figure 2 so both contain numerical values.
Comments 5:
Lines 231 – 232. You mention the use of the internal mammary for CABG. Do you not use this technique and if so, how many patients have had this procedure?
Response 5: Thank you for pointing this out. When discussing pleural effusion after CPB, we describe internal mammary artery harvesting as essential. Unfortunately, in our data, when talking about CABG surgery with or without the usage of the mammary artery, nether graft count was not available as an asset due to data collection specification.
Comments 6: The cause of hypoalbuminemia is not discussed... it is possible to appear in those with a more severe inflammatory reaction after CPB or due to a large volume of administered fluids. You don't mention what were the average CPB time and also the relationship between CPB duration and complications, being known that the longer the CPB duration is, the greater the inflammatory response will be. Please enter the CPB duration values, assess the association between CPB time and PPC and after that redo the statistics. How did you monitor the volume status in the postoperative period? (Vexus score?)
Response 6: Agre. The causes of hypoalbuminemia after CPB were not mentioned. I have modified the discussion section (lines 249 – 252) by mentioning some possible causes of early hypoalbuminemia after CPB. CPB time (min) and CPB time are mentioned in Table 5, and Table 4A has been added to the Appendix. Since CPB time, CPB priming volume, or fluid balance after surgery did not show statistical significance in the primary analysis; these values were not further analyzed. However, I agree that in the context of a possible multifactorial cause of pleural effusion, fluid management should be an important factor considered.
In our center, volume status is assessed using a combination of the patient's clinical conditions (e.g., hypotension), central venous pressure, urine output, and transthoracic echo findings.
Comments 7: Line 254. lack instead of Lack
Response 7: I have revised this point. Thank You.
Comments 8: Lines 263 – 264. Are you sure that EF was not also measured postoperatively? It is hard for me to believe that in a reference university center the cardiac function of patients undergoing cardiac interventions is not monitored in the postoperative period. If this is really true, you better not mention it in the final text.
Response 8: Thank you for pointing this out. I have revised the discussion section related to this point (lines 268 - 270). In our research, ejection fraction (EF) before surgery did not significantly affect pleural effusion development, but data about EF after surgery were not analyzed due to its lack of availability. Compromised left ventricular ejection fraction after surgery could impact early pleural effusion development. Unfortunately, our dataset did not allow us to analyze it.
Reviewer 3 Report
Comments and Suggestions for Authors
Dear authors, thanks for the opportunity to revise your paper entitled "Identifying Early Risk Factors for Postoperative Pulmonary Complications in Cardiac Surgery Patients". In this study authors tried to understand factors associated to the development of different postoperative pulmonary complications after cardiac surgery. Authors have done a great job and the study is well conducted and can be interesting for readers. Anyway, there are some concerns that should be addressed before publication:
- Authors should discuss in the introduction and in the discussion part the role of postoperative thoracic pain due to sternotomy in limiting thoracic excursions and limiting respiratory function. in this regard, please add this important recent reference of how sternotomy pain could affect respiratory performance: 10.1136/rapm-2024-105430
- Who waived the need for informed consent to partecipate to the study? please specify.
- How were the patient ventilated in the ICU? Did they switch to pressure support before awakening?
- Did authors recorded and analysed data on drug administration in the ICU? Correlation with PPC?
- Also extubation time..data? correlation?
Comments on the Quality of English LanguageMinor editing of English language required
Author Response
Thank you very much for taking the time to review our manuscript. Your comments were very useful in improving our research paper. Please find the detailed responses below and corrections in the re-submitted files.
Comments 1: Authors should discuss in the introduction and in the discussion part the role of postoperative thoracic pain due to sternotomy in limiting thoracic excursions and limiting respiratory function. in this regard, please add this important recent reference of how sternotomy pain could affect respiratory performance: 10.1136/rapm-2024-105430
Response 1: Thank you for pointing this out. Postoperative pain management after cardiac surgery is an important topic that covers postoperative pulmonary complications, especially atelectasis and pneumonia. Our center provides a parasternal block as a standard for postoperative pain management after heart surgery. Unfortunately, pain management was not assessed as a factor for postoperative pulmonary complications in our study. Based on your suggestion, we have added some additional references regarding pain control and respiratory function. (lines 296 – 298)
Comments 2: - Who waived the need for informed consent to partecipate to the study? please specify.
Response 2: Individual consent for this analysis was waived since data was collected retrospectively. The study received approval from Riga Stradins University medical ethics committee.
Comments 3: - How were the patient ventilated in the ICU? Did they switch to pressure support before awakening?
Response 3: I modified this point from materials and methods section (lines 94 – 98). Patients were mechanically ventilated according to the local protocol using pressure regulated volume controlled mode, FiO2 - 0.6, 6 ml/kg tidal volume, and PEEP set at 5 mmHg. Prior to extubation, patients were ventilated in pressure support mode. After extubation, oxygen was administered via a non-rebreathing face mask at a flow rate of 10 l/min.
Comments 4: - Did authors recorded and analysed data on drug administration in the ICU? Correlation with PPC?
Response 4: We analyzed vasopressor and inotrope usage in the ICU after surgery, but no statistical significance was found. A statistical analysis can be found in Table A4, added in the appendix. No other drugs (e.g., sedatives, opioids) were analyzed.
Round 2
Reviewer 2 Report
Comments and Suggestions for Authors
Dear Authors, thank you for the changes you did in the manuscript.
I ask you to insert the values of CPB time, presented in the appendix, in the final form of Discussion section and please make a comment about the relation between CPB time-inflammation-pleural effussion.
Author Response
Thank you for your comment.
Comments 1: I ask you to insert the values of CPB time, presented in the appendix, in the final form of Discussion section and please make a comment about the relation between CPB time-inflammation-pleural effussion.
Response 1: I have modified the “Discussion” paragraph by comparing CPB time for patients with and without left-sided pleural effusion. I tried to point out the possible impact of CPB causing a systemic inflammatory response, fluid accumulation, and hemodilution. Changes can be found – in the "Discussion" paragraph, lines 320 - 327.
Reviewer 3 Report
Comments and Suggestions for Authors
Authors have successfully addressed all the comments raised. The quality of paper is improved after revision. Thanks again for the opportunity to revise your paper.
Comments on the Quality of English LanguageMinor editing of English language required.
Author Response
Thank you very much for your time and revision.